# Effects of Feed Additives (*Nannochloropsis gaditana* and *Hermetia illucens*) on Growth and Expression of Antioxidant and Cytokine Genes in Nile Tilapia (*Oreochromis niloticus*) Subjected to Air Exposure Stress

**DOI:** 10.3390/ani15121776

**Published:** 2025-06-17

**Authors:** László Ardó, Zsuzsanna J. Sándor, Márton Orbán, János Szakáli, Janka Biró, Anita Annamária Szűcs, Gyula Kovács, Michelle Lévai, Balázs Gregosits, Zsuzsanna Brlás-Molnár, Emese Békefi

**Affiliations:** 1Research Centre for Fisheries and Aquaculture, Hungarian University of Agricultural and Life Sciences (MATE), Anna Liget. u. 35, 5540 Szarvas, Hungary; ardo.laszlo@uni-mate.hu (L.A.); nagyne.biro.janka@uni-mate.hu (J.B.); szucs.anita.annamaria@uni-mate.hu (A.A.S.); kovacs.gyula@uni-mate.hu (G.K.); brlas-molnar.zsuzsanna@uni-mate.hu (Z.B.-M.); bozanne.bekefi.emese@uni-mate.hu (E.B.); 2Vitafort First Hungarian Feed Production and Distribution Zrt., Szabadság u. 3., 2370 Dabas, Hungary; orban.m@vitafortasia.com (M.O.); gregosits.b@vitafort.hu (B.G.); 3Vitafort Agro Asia Zrt., Szabadság u. 3., 2370 Dabas, Hungary; szakali.vaa@gmail.com; 4ADC Aquatic Development Company Ltd., AngNamHoum Village, Naxaything District, Vientiane 0114, Laos; flevai@aranyponty.hu

**Keywords:** stress, additives, *Hermetia illucens*, *Nannochloropsis*, gene expression

## Abstract

Over the past 20 years, the production of Nile tilapia (*Oreochromis niloticus*) has grown rapidly in Laos, accounting for over 38% of the country’s total finfish aquaculture production. With the intensification of aquaculture, there has been increasing interest in the use of plants, herbs, and algae as immunostimulants in fish diets. In this study, the impact of different feed additives was examined on juvenile fish reared in cages over a 7-week feeding period. The fish were fed with four different types of diets and subsequently subjected to air exposure. Overall, both feed additives supported the fish in coping with air exposure stress during farming.

## 1. Introduction

Although Laos is a landlocked country, it possesses abundant freshwater resources that are well-suited for fish farming. However, its geographical location significantly differentiates the structure of its fisheries and aquaculture from that of neighboring countries with access to the sea. Capture fisheries and aquaculture are of great importance in the employment of rural communities and nutrition of the people living in the countryside. Annual fish consumption per capita exceeds 25 kg, which makes Laos one of the world’s leading fish-consuming countries. For a long time, carps (common carp, Chinese carps, and Indian major carps) were dominant in Lao fish production, but in the last 20 years, the production of Nile tilapia (*Oreochromis niloticus*) has developed rapidly. Tilapia production reached 55,260 tons in 2022, which makes up 38% of the total finfish aquaculture production based on the dataset from FAO FishStatJ [1].

In parallel with the expansion and intensification of the aquaculture sector in Laos, it has been highlighted that the feed sector requires more specialized functional feeds to protect fish from diseases. Functional feeds in aquaculture are specially formulated diets designed not only to meet the basic nutritional requirements of aquatic species but also to promote health, enhance growth performance, and improve resistance to stress and diseases. These feeds often include additives such as probiotics, prebiotics, immunostimulants, plant extracts, carotenoids, β-glucans, or essential fatty acids. By supporting the immune system and overall well-being of farmed species, functional feeds contribute to more sustainable and productive aquaculture practices [2,3]. Microalgae of the genus *Nannochloropsis* are characterized by a high content of polyunsaturated fatty acids (PUFAs), carotenoids, vitamins, and polyphenols [4], besides their advantageous protein (22.2–45%) and lipid content (15.1–45%) [5,6]. Several studies focused on its impact on the non-specific immune system of Nile tilapia. For instance, dietary supplementation of *Nannochloropsis oculata* beneficially affected the innate immune parameters and immune-related gene expression [7]. These effects were also observable following an air exposure stress. In further studies, the enhancement of growth performance, feed utilization, serum biochemical indices, antioxidant activities, and immune response of Nile tilapia due to the *N. oculata* supplementation was also reported [8,9]. Similar positive effects of *N. gaditana* inclusion in the diet were described in gilthead seabream [10,11].

Based on the above-mentioned promising characteristics of *Nannochlorophsis* genus, in the frame of iFishienci project it was aimed to select several strains of various biomass sources at Norwegian Research Centre (photoautotrophic microalgae, fungi, heterotrophic microalgae) regarding their nutritional values (especially protein, total fatty acid, and omega-3 fatty acid content) or antioxidant capacities and produce it at high scale for dietary inclusion for animals. Finally, biomass (more than 17 kg dry weight) of photoautotrophic microalgae *Nannochloropsis gaditana* CCMP526 was produced and partly utilized in the preparation of the experimental feed EXP-A.

Black soldier fly larvae (BSFL) *(Hermetia illucens*) meal has a good nutritional profile in terms of vitamins, minerals, nucleotides [12], and bioactive compounds [13]. BSFL meal supplementation positively affected the health status of fish in several species, such as rainbow trout (*Oncorhynchus mykiss*) [14], European seabass (*Dicentrarchus labrax*) [15], yellow catfish (*Pelteobagrus fulvidraco*) [16], and marron (*Cherrax cainii*) [17]. In Nile tilapia, BSFL meals stimulated innate immunity by increasing lysozyme activity [18,19]. Enhanced immunity due to improved lysozyme and peroxidase activities in the skin mucus of Nile tilapia fed 4% and 6% BSFL meal was also observed [20]. However, except for the above-mentioned publications, only a few studies focused on the effect of BSFL as an immunostimulant in the case of Nile tilapia.

Our primary aim was to emphasize the importance of using formulated feeds supplemented with additives to boost aquaculture productivity. According to this, the objective of this study was to compare the effects of dietary inclusion of *N. gaditana* meal CCMP256, defatted black soldier fly larvae meal, and a mix of two commercially available feed additives on production traits, antioxidant defense system, innate immune response, and stress tolerance of intensively reared Nile tilapia. In order to achieve this goal, juvenile fish were fed with different diets, and finally, an air exposure stress was administered.

## 2. Materials and Methods

### 2.1. Experimental Design and Fish Husbandry

The location and infrastructure for the implementation of the experiment were provided by Aquatic Development Company in Vientiane, Laos, where 1000 individuals of a monosex batch of Nile tilapia hybrids produced by the combination of GIFT (Genetically Improved Farmed Tilapia) and Big Nin strains, with approx. 100 g body weight was set up for the feeding trial. The 2-week-long habituation period started on 20 August 2021. Two cage systems were installed in the same lake, and each system consisted of four cages with a 45 m^3^ volume per cage. The water supply of the lake was provided by the Namhoum reservoir. Fish stock was allocated into eight groups; the treatments were applied in duplicates. According to the experimental design, 120 fish were accommodated in one unit (Figure 1). The stocking density was set lower than that typically used in commercial conditions to promote intensive weight gain. Furthermore, the limited availability of the tested *N. gaditana* meal imposed constraints on feed production and decreased the number of fish involved in the nutritional trial.

The feeding trial lasted 7 weeks and ended on 22 October 2021. On the second week of the habituation, as part of the iFishienci project activity, the iBOSS system (with the new sensor) was installed by the online support of Bioceanor (Valbonne, France). The following parameters were recorded by the instrument to the server every 20 min: O_2_ level, pH, temperature, and conductivity. The O_2_ saturation fluctuated between 41.5 and 66.6% with an average value of 55.9 ± 9.6%. The pH level was 6.2 ± 0.31, while the temperature was almost constant at 31.3 ± 0.6 °C. For the conductivity of water, 34.1 ± 0.63 µS/cm was measured. The feed was offered twice a day, and the daily ratio was set to 3.0% of total biomass in the first week, with the ratio decreasing during the trial according to the growth of fish (Table 1). To determine the weekly weight gain and to check the general health conditions of the population, the fish were measured on a weekly basis. During the sampling, the length and weight of fish (15 fish/cage) were recorded by every feeding group to provide data for the calculation of production performance indicators.

### 2.2. Ethical Issues

All procedures involving fish were conducted in accordance with the Law on Animal Husbandry and Veterinary (Revised) No. 08/NA, dated 11 November 2016, of the Lao People’s Democratic Republic. All efforts were made to minimize the suffering of the fish.

### 2.3. Feed Preparation

Three experimental diets were formulated as follows: treatment EXP-A, feed supplemented with 3.5% *N. gaditana*; treatment EXP-I, feed supplemented with 3.5% black soldier fly larvae meal; treatment EXP-S, feed supplemented with bioactive compounds in 0.4%. The last one was a mixture of commercially available feed additives, Yang (Lallemand, France) and Syrena Boost (Delacon Biotechnik, Engerwitzdorf, Austria), possessing high technological adaptability and good physiological effects. By this construction the performance of the fish fed with algae and insect protein supplemented feeds could be compared not just with the commercial feed used for Nile tilapia (treatment COM) but with a group offered a feed containing commercially available bioactive additives. The experimental diets were set to iso-nitrogenous and iso-energetic. The feeds were produced by Nongteng feed mill (Chansavang Village, Sikhottabong District, Vientiane Capital, Laos). The formulation, proximate, and calculated chemical composition of the diets are presented in Table 2.

### 2.4. Chemical Analysis of the Diets

The chemical composition of the feeds was analyzed by standard Hungarian and AOAC methods as follows: the crude protein was determined with the Dumas method based on Hungarian MSZ EN ISO 16634-1:2009 [21] regulation using Rapid N Cube nitrogen/protein analyzer (Elementar Analysensysteme GmbH, Langenselbold, Germany). The crude fat was determined according to the AOAC 945.16 based on the Soxhlet method using an automatic system (SOXTHERM^®^ Unit SOX416, Gerhardt, Germany) and diethyl ether (boiling point, 40–60 °C) as a solvent. For Ca and P determination, the Hungarian standard method MTK 2004 III. 12. was applied, when the samples were incinerated at 550 °C, the residual ash was digested in 10% hydrochloric acid and used after dilution to set the calibration range. The measurements were performed on an inductively coupled plasma optical emission spectrometer (iCAP 7400, Thermo Fischer Scientific, Waltham, MA, USA).

### 2.5. Sampling and Air Exposure Stress

At the end of the 7-week feeding experiment trial, expression of genes involved in growth, non-specific immune response and antioxidant defense system of the fish were studied. For this reason, liver and head kidney samples were collected from six fish per treatment (24 individuals altogether). Fish after 24 h of feed deprivation were anesthetized with an overdose of clove oil, and 100 mg of head kidney and liver samples were placed into 1 mL of RNAlater reagent (Invitrogen, Thermo Fisher Scientific, Waltham, MA, USA) for 1 day at 4 °C, followed by storage at −20 °C. After the feeding experiment, six fish per treatment (24 individuals altogether) were netted out of the cages and exposed to air for 5 min. There is no referred, specific protocol for conducting air exposure stress experiments with tilapias. Therefore, the exposure time was selected according to data in the literature. In a similar experiment [7], 1 min was applied, and 5 min was regarded as adequate, because tilapias can survive on air for much longer times [22,23]. After this period, all 24 stressed fish were immediately put back into water and over-anesthetized with clove oil. Head kidney and liver samples were taken from all 24 fish. The detailed structure of the feeding and stress experiment is presented in Figure 1.

### 2.6. Growth Parameters

In order to determine the growth performance and nutrient utilization of the fish, the following parameters were measured and calculated at the end of the trial:Specific Growth Rate (g/day), SGR = 100 × (ln FBW − ln IBW)/tFBW = final average weight at the end of the experiment (g)IBW = initial average weight at the beginning of the experiment (g)t = experimental time in daysWeight gain (g), WG = FBW − IBWCondition factor (g cm^−3^), CF = 100 × FBW/L^3^L—total body length at the end of the trialFeed conversion ratio (g/g), FCR = total amount of feed given (g)/final total body weight − initial total body weight (g)Specific Feed Cost (€kg^−1^ fish) = feed cost × FCR

### 2.7. Analysis of Gene Expression

The liver and head kidney samples were transported to the laboratories of the Research Institute of Fisheries and Aquaculture of MATE, Hungary, in order to perform the analysis. During the transport, the samples were kept below 4 °C, and after arriving at the laboratory site, they were stored at −20 °C until the analysis. Total RNA was isolated from the samples using the Aurum Total RNA Mini kit (Bio-Rad, Hercules, CA, USA), according to the manufacturer’s instructions. Concentration and purity of RNA were measured using a NanoDrop 2000 spectrophotometer (Thermo Scientific, Waltham, MA, USA), while the quality of the RNA was checked with agarose gel electrophoresis. A total of 400 ng of mRNA from each sample was reverse transcribed into cDNA using the qPCRBIO cDNA Synthesis kit (PCR Biosystems, London, UK). cDNAs were analyzed in real-time quantitative PCR (qPCR) reactions. These reactions were run in triplicate, using a LightCycler 96 instrument (Roche, Basel, Switzerland) and the Luna Universal qPCR Master Mix (New England Biolabs, Ipswich, MA, USA). Expression of genes related to oxidative stress response (*sod*, *cat*, *gpx*) and growth (*igf-1*) was measured from liver samples, whereas expression of genes related to innate immune response (*tnf-α*, *il-1β*, *il-8*, *ifn-γ*) was measured from head kidney samples. *18S rRNA* was used as a reference gene. The primers used are presented in Table 3. The qPCR reaction was carried out in a final volume of 20 µL consisting of 10 µL of master mix (2×), 1 µL of each primer (10 µM), 5 µL of cDNA (reverse transcription reaction mix), and 3 µL of nuclease-free water. The thermal profile for all reactions was 95 °C for 10 min, followed by 45 cycles at 95 °C for 15 s, and 60 °C for 30 s. The specificity of the reactions was checked by melting curve analysis, and no mispriming or primer dimers were found. The mean threshold cycle (Ct) values were calculated, and the qPCR data were analyzed by the 2^−ΔΔCt^ method described by Livak and Schmittgen [24]. The efficiencies of qPCR reactions were determined using standard curves, and serial dilutions were made from cDNAs of a liver and a head kidney sample. These cDNAs were diluted to 10×, 30×, 90×, 270×, and 810×. Quantitative PCR reactions were carried out on these dilutions with all primer pairs in triplicate. Standard curves were drawn for each primer pair by plotting Ct values against the log10 of different dilutions of cDNA sample solutions. Efficiencies (E) were calculated from the slopes of the standard curves using the equation E = 10(−1/slope), and the results are shown in Table 3.

### 2.8. Statistical Analysis

Growth data were statistically processed by one-way analysis of variance (ANOVA) followed by Tukey Multiple Comparison Test (*p* < 0.05) using the SPSS software version 22 (IBM Corp, Armonk, NY, USA). The significant differences were considered at *p* < 0.05. The Kolmogorov-Smirnov test was used to assess normality. The homogeneity of the variances was checked by the Tukey HSD test. Gene expression data were analyzed and figures were drawn using SigmaPlot version 12.0 (Systat Software). Normality of data and homogeneity of variances were checked using the Shapiro-Wilk and Tukey HSD tests, respectively. Differences between experimental groups before and after air exposure were analyzed by one-way ANOVA, followed by Student–Newman–Keuls test, whereas an independent samples *t*-test was used to determine the differences between samples taken before and after the air exposure in each experimental group. In both cases, the differences were considered significant at *p* < 0.05.

## 3. Results

### 3.1. Growth Performance

The growth performance of Nile tilapia is summarized in Table 4. During the 7-week trial, the fish increased their body weight by almost 2.5 times with a specific growth rate (SGR) between 1.67 and 1.84 g day^−1^. Feed conversion rate varied between 1.29 and 1.57. The detected mortality was not higher than 7.1% (the survival rate was higher than 92.9%). According to the statistical analysis performed, no significant differences were detected among any of the treatments in the production and nutrient utilization parameters, survival rate, and condition factor of the fish. The specific feed cost (feed cost per unit of weight gain) calculated for each treatment showed a slight increase with the inclusion of additives, with the highest cost observed in the diet supplemented with *N. gaditana*.

### 3.2. Expression of Genes Related to Growth, Stress, and Immune Response

At the end of the feeding experiment, the expression levels of genes involved in the antioxidant defense system *(sod*, *cat*, *gpx*) significantly (*p* < 0.05) increased in liver samples of group EXP-A, compared to the other groups (Figure 2a–c). However, the expression of *igf-1*, a gene controlling growth, was not significantly different among the groups (Figure 2d). In the head kidney samples, significant (*p* < 0.05) increases were detected in the expression levels *il-8* and *ifn-γ* in all treated groups compared to the group fed with commercial diet (Figure 3c,d).

Following the 5-min air exposure stress, expression levels of *sod*, *cat*, and *gpx* in liver samples were significantly (*p* < 0.05) higher in group fed with *N. gaditana* (EXP-A) compared to the other groups (Figure 2e–g), whereas the expression of *igf-1* did not differ among the groups (Figure 2h), similarly to the samples taken from before stress (Figure 2a–d). There were no significant (*p* < 0.05) differences within treatments in the expression levels of genes related to the immune response after air exposure in the head kidney samples (Figure 3e–h).

## 4. Discussion

Feed additives have been added to fish diets in order to protect fish from diseases or minimize other possible risks during the culturing period or transportation events. We also thought that some feed additives had a protective effect against oxidative stress by reducing the formation of free radicals. It is well known that the intake of antioxidants and polyphenolic compounds from microalgae positively affects the antioxidant defense systems of fish [28]. BSFL meal was considered to interact with the non-specific immune and antioxidant defense system due to its significant chitin content [18,19]. According to the manufacturer’s technical report, Syrena Boost at a 200 mg/kg level can enhance the growth and survival of Nile tilapia fingerlings. Parallel with this, Yang was added in 1 g/kg doses, which was documented as having positive effects on pathogen binding, skin mucus production, and intestinal integrity in different aquatic species [29]. Based on these findings, we hypothesized that this product should have an immunoboosting effect. When comparing the three experimental treatment groups, we found that all diets exhibited similar production parameters during the experimental period. These findings support the assumption that *N. gaditana* and black soldier fly larvae meal are digestible feed additives for Nile tilapia without compromising the growth traits of the species. It could be concluded that all the supplementation materials contributed to good production traits of tilapia juveniles. However, some dietary disadvantages could be observed regarding the final body weight, specific growth rate, and feed conversion rate in the commercial group, whose diet formulation differed from the rest of the diets in composition. As shown in Table 1, the diets were isonitrogenous and isoenergetic, but they contained different ratios of fish meal and soybean. The experimental diets were formulated with 51–53% fish meal content, compared to the commercial diet’s 40%. Consequently, the soybean meal ratio is significantly higher in the commercial diet (36%) compared to experimental diets (approximately 18%). It appears that the lower level of fish meal in this diet, along with the higher soybean content, adversely affected the growth of the fish and their feed utilization. This trend is clearly visible despite the large variance observed in these parameters among parallel cages. We concluded that the variations in the fishmeal-to-soybean meal ratio are reflected less in amino acid composition and more in differences in nutrient digestibility caused by the soybean. On the other side, the feeds were supplemented in 3% with premix containing synthetic amino acids to satisfy the requirement.

Superoxide dismutase (SOD), catalase (CAT), and glutathione peroxidase (GPX) enzymes are parts of the antioxidant defense system, which protects the cells from peroxidation of lipids, proteins, and DNA by eliminating reactive oxygen species (ROS), which appear during stressful conditions or the activation of innate immune system [30]. In our experiment, feeding the fish with *N. gaditana* significantly enhanced the expression of these three antioxidant enzymes in the liver. This result is consistent with previous studies, in which the expression of these genes increased by feeding tilapias with *N. oculata*, *Golenkinia longispicula*, or a mixture of different microalgae [9,31,32]. Based on our findings of upregulation of *cat*, *sod*, and *gpx* genes in the liver, it seems that this additive could activate the antioxidant functions of juvenile GIFT. Microalgae of the genus *Nannochloropsis* are rich in valuable carotenoids like violaxanthin, zeaxanthin, astaxanthin, or β-carotene [4]. These molecules can neutralize ROS by accepting electrons to their conjugated double bonds [33]. Beside this direct effect, they can activate the transcription factor Nrf2 (Nuclear factor erythroid 2-related factor 2), which increases the expression of genes *sod*, *cat*, and *gpx* [34,35]. This explains the significant increase in the expression of these genes measured in group EXP-A. A slight increase in antioxidant gene expression in EXP-S and EXP-I groups without significant differences from the commercial group was observed. In this sense, the additives administered to EXP-S and EXP-I diets were not strong enough to have an impact on the antioxidant defense system of the studied fish individuals. On the other hand, the lack of antioxidative effect of EXP-S diet could be explained by the low inclusion level administered into the feed (0.4%). The homogenization of solid particles in such low concentrations into the feed may lead to an uneven distribution, resulting in inconsistent amounts being delivered to the fish. Similarly, the dietary inclusion of BSFL at a dose of 3% does not appear to have generated an antioxidant effect compared to higher doses reported in the literature [36].

Insulin-like growth factor 1 (IGF-1) mediates the effect of growth hormone (GH) on skeletal muscle. The main target organ of GH is the liver, where the transcription of the *igf-1* gene is activated via various signaling pathways [37,38]. Synthesis and release of hepatic IGF-1 is influenced by the nutritional status of fish [26,37]. In our study, none of the experimental diets changed the expression of *igf-1* gene in the liver, which indicates that these diets did not have significant effects on the nutritional status, and this observation is in correlation with our growth performance data.

Pro-inflammatory cytokines play a key role in initiating and mediating the innate (non-specific) immune response by activating their cellular components (macrophages, natural killer cells, and granulocytes) [39]. In our experiment, the expression of four pro-inflammatory cytokines (*tnf-α*, *il-1β*, *il-8*, and *ifn-γ*) was measured in the head kidney, which is one of the main lymphoid organs of fishes [40]. Two of them (*il-8* and *inf-γ*) had significantly increased expression levels after 7 weeks of feeding with all three experimental feeds, compared to the control group. Feed additives used in our study contain various compounds, as was mentioned before, which may act as immunostimulants in order to enhance the innate immune response. Similarly, in the EXP-S diet, a combination of phytogenic and yeast-based feed additives was added, which contains saponins and glucans, compounds with immunostimulating activities. Similar results were reported in hepatic gene expression of Nile tilapia after 90 days of feeding with different doses of the microalga *Golenkinia longispicula* [32], after 7 weeks of feeding with *N. gaditana* in 5 or 10% dose [9], and 30 days of feeding with *Spirulina platensis* in 0.25 or 0.5% dose [41]. β-carotene and phycocyanin extracted from *Spirulina platensis* also increased the expression of *il-1β* and *ifn-γ* after 70 days of feeding [42]. However, a mixture of *Spirulina* and *Schizochytrium* species and *N. oculata* in 1.5% and 3% doses significantly decreased the expression of *il-1β* and *tnf-α* in Nile tilapia spleen [31]. *N. oculata* in 5% or 10% doses reduced the expression of *il-1β* in the liver and intestine, and the expression of *tnf-α* in the intestine, but not in the liver [7]. These results indicate that the effect of microalgae on the expression of pro-inflammatory cytokines is highly dependent on the tissue and the microalgal species. Similarly, a combination of black soldier fly larvae meal and chitinase significantly enhanced the expression of *il-6* in the head kidney of Nile tilapia juveniles after 53 days of feeding. However, no BSFL meal and chitinase combinations had a significant effect on the expression of *il-1β* in head kidney compared to the control or the expression of both cytokines in spleen, suggesting that BSFL meal can have a similar effect as microalgal species [19]. A phytogenic feed additive, pineapple peel powder, in 5, 10, and 20 g/kg doses significantly increased *il-1β* and *il-8* expression in Nile tilapia liver after 8 weeks of feeding, showing its immunomodulatory effect [27]. Fish feed supplemented with 500 mg/kg β-glucan significantly enhanced the expression of *il-1β* in Nile tilapia liver after 60 days of feeding [43]. In another experiment, the same dose of β-glucan increased the expression of *il-8* in the liver of Nile tilapias after 30 days [44]. These latter results with β-glucan indicate that the exact effect of immunostimulants on the expression of pro-inflammatory cytokines depends on the feeding regime as well. Accordingly, our data provide evidence that all the tested additives may help enhance the non-specific immune system of the fish after the feeding period.

Following the stress experiment, the EXP-A group maintained its increased antioxidative capacity during the stress event compared to the rest of the treatments, similar to what was observed at the end of the feeding period. The expressions of antioxidant enzyme genes (*sod*, *cat*, *gpx*) and *igf-1* were very similar before and after air exposure. Only the expression of *gpx* in the EXP-S group increased significantly following the stress. In an experiment with Nile tilapia [8], the expression of *gpx* in the liver was higher or lower following air exposure, compared to the pre-stress levels, depending on the dose of *N. oculata* mixed into the feed (5 or 10%, respectively). These doses were higher than the 3.5% of *N. gaditana*, which is the inclusion level we had in our experiment, demonstrating that the antioxidant effect of microalgae is dose-dependent. Feeding Nile tilapias with 500 mg/kg β-glucan supplementation for 30 days significantly increased the expression of *gpx* in the liver of fish exposed to a subacute dose of deltamethrin, compared to the non-exposed fish [43]. This was similar to what we found in the EXP-S group after the air exposure. However, in this case, the expression of *cat* was also significantly higher in the stressed group, which was different from our result. This indicates that the type (acute or subacute) of stress also affects the expression of antioxidant genes in Nile tilapia liver.

The expression of pro-inflammatory cytokine genes (*tnf-α*, *il-1β*, *il-8*, *ifn-γ*) detected in the head kidney after the air exposure did not differ significantly among the experimental groups in our study, which indicates that the acute stress decreased the immunomodulatory effect of the experimental feeds detected at the end of feeding. However, a positive trend in the non-specific immune response of tilapias fed with feed additives compared to the control group could be observed following air exposure. Compared to the pre-stress state, only the expression of *ifn-γ* decreased significantly in the EXP-A group (Figure 3d,h). Similar results were found in Nile tilapia liver and intestine, where the expression of *tnf-*α and *il-1β* after air exposure was not different between the experimental groups, which were fed with feeds supplemented with 5 and 10% of *N. oculata* [7]. In the same study, the expression of *tnf-α* significantly decreased after the stress, compared to the pre-stress level in the 10% group. In contrast to this and to our results, expression of *il-1β* remained elevated in the liver of Nile tilapias fed with 0.25 and 0.5% *Spirulina platensis* following acute cold or hypoxia stress [41]. However, this can be explained by the stronger effect of direct air exposure as a stressor. Following 30 days of exposure to a subacute dose of deltamethrin, the expression of pro-inflammatory cytokines either increased (*ifn-γ*), decreased (*il-8*), or remained at the same level (*il-1β*) in Nile tilapia liver, compared to the control. In the same experiment, feeding the deltamethrin-treated fish with 500 mg/kg β-glucan during the experiment significantly increased the expression of *il-8* and *il-1β*, compared to the fish treated with deltamethrin only [43]. These results suggest that the expression of pro-inflammatory cytokines depends on the type of stress, similar to antioxidant genes. Finally, we concluded that all three studied feed additives in their respective doses improved the innate immune response of Nile tilapias after 7 weeks of feeding. Moreover, the elevated antioxidant capacity of the *N. gaditana*-fed group was maintained even after the stress.

## 5. Conclusions

Following the 7-week feeding trial, all supplemented diets performed similarly, with good production and nutrient utilization parameters, and feed cost per unit of weight gain. It can be assumed that *N. gaditana* supplementation enhanced the antioxidant defense system of the fish by increasing the expression of genes encoding important antioxidant enzymes. Furthermore, all tested additives enhanced the non-specific immune response of the fish after the feeding period. Following air exposure, the increased antioxidant effect was maintained in the fish fed with *N. gaditana* supplementation, while the immunomodulatory effect of the additives observed following feeding was somehow decreased via air exposure. Based on our results, we recommend the application of any of the studied additives in the tested doses to enhance the innate immune response of Nile tilapia. In addition to its immunomodulatory properties, *N. gaditana* in 3.5% can be applied to increase the antioxidant defence capacity as well.

## Figures and Tables

**Figure 1 animals-15-01776-f001:**
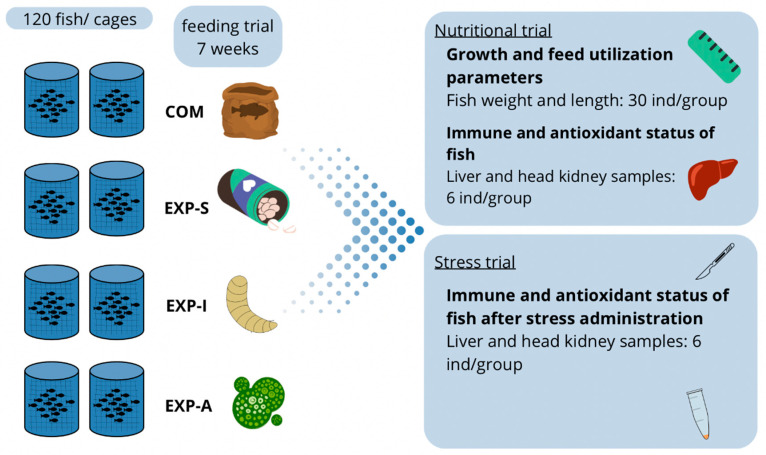
Experimental and sampling design. COM: Nile tilapia commercial feed (control group); EXP-S: experimental feed supplemented with a mixture of two commercially available feed additives (Yang and Syrena Boost) in 0.4%; EXP-I: experimental feed supplemented with 3.5% black soldier fly larvae meal; EXP-A: experimental feed supplemented with 3.5% *N. gaditana*.

**Figure 2 animals-15-01776-f002:**
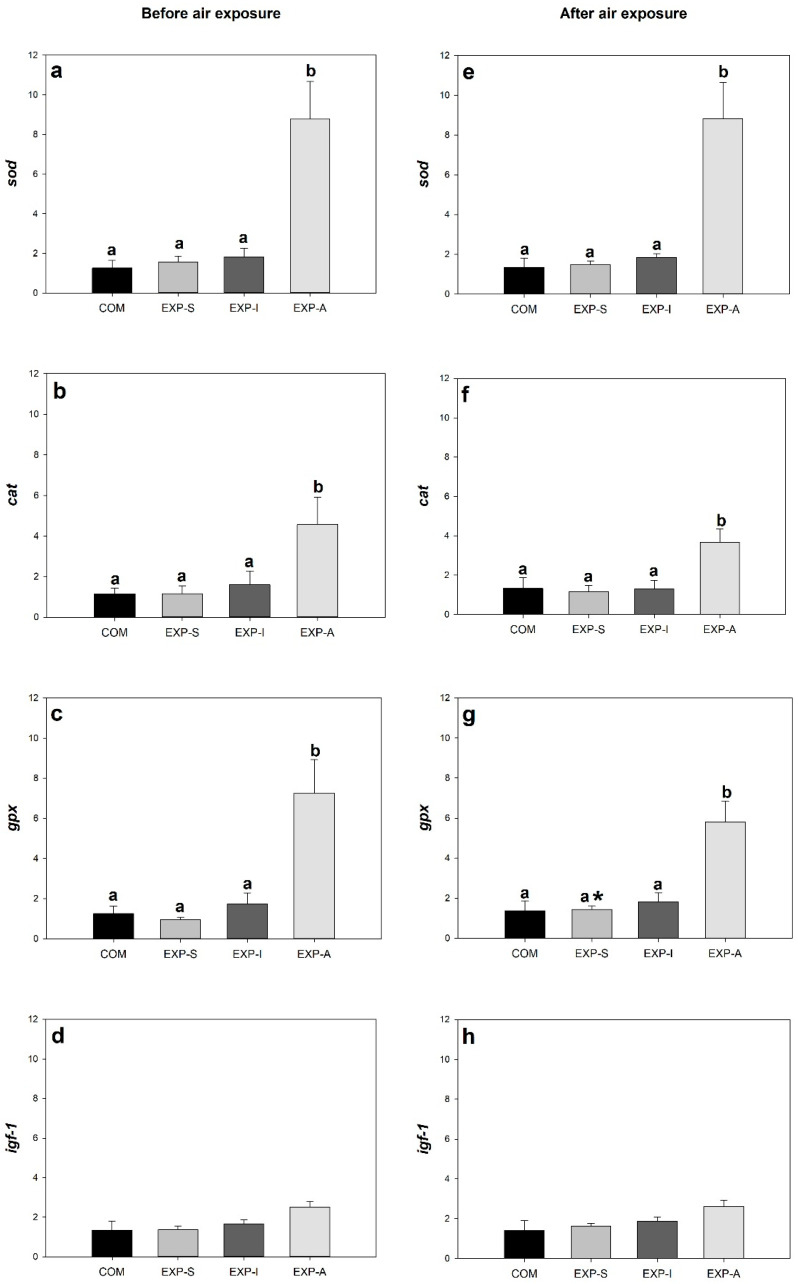
Relative expression of genes involved in growth and antioxidant defense system in the liver of the fish before (**a**–**d**) and after air exposure stress (**e**–**h**). Different letters mean significant (*p* < 0.05) differences between the groups. Asterisk (*) means a significant (*p* < 0.05) difference between values obtained before and after air exposure in the same group. *sod*: superoxide dismutase; *cat*: catalase; *gpx*: glutathione peroxidase; *igf-1*: insulin-like growth factor 1.

**Figure 3 animals-15-01776-f003:**
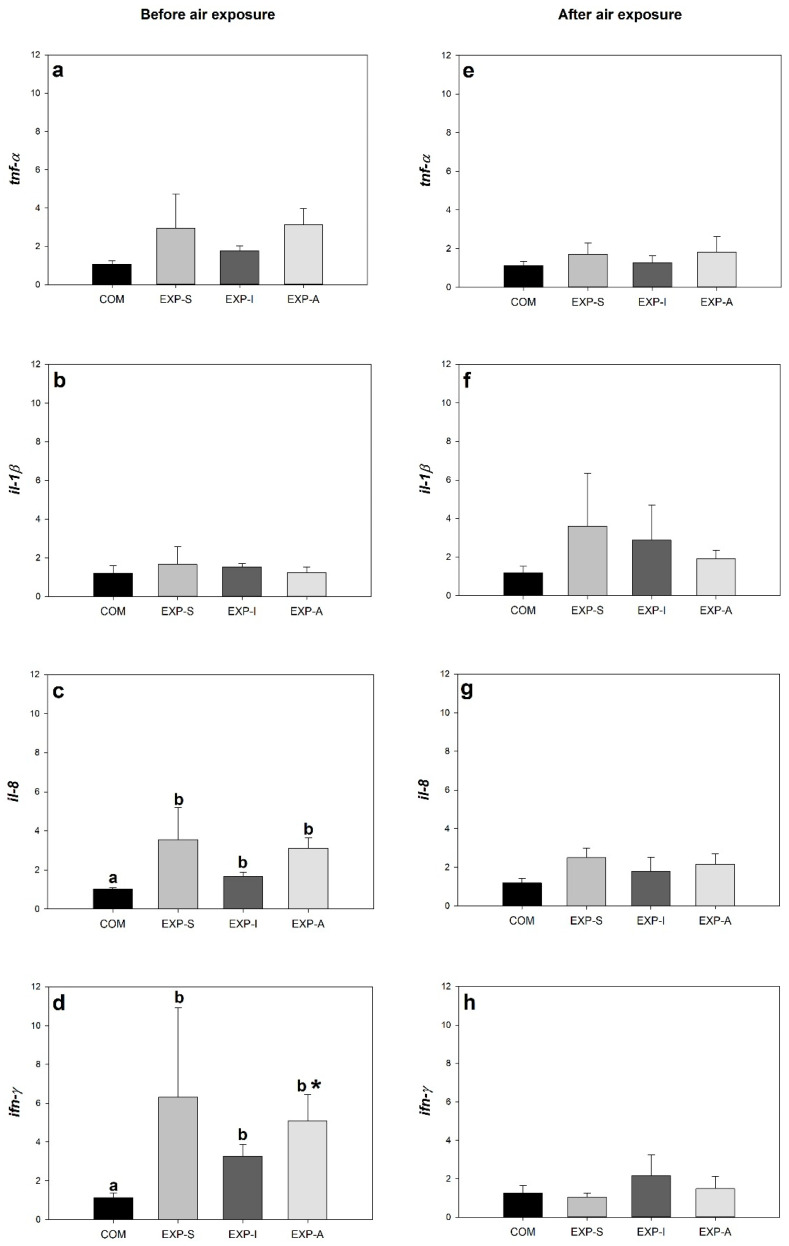
Relative expression of genes involved in non-specific immune response in the head kidney of the fish before (**a**–**d**) and after air exposure stress (**e**–**h**). Different letters mean significant (*p* < 0.05) differences between the groups. Asterisk (*) means a significant (*p* < 0.05) difference between values obtained before and after air exposure in the same group. *tnf-α*: tumor necrosis factor alpha; *il-1β*: interleukin-1 beta; *il-8*: interleukin-8; *ifn-γ*: interferon gamma.

**Table 1 animals-15-01776-t001:** Feeding rate (% of total biomass) during the trial.

Weeks	1.	2.	3.	4.	5.	6.	7.
feeding rate %	3	2.5	2.5	2	2	2	2

**Table 2 animals-15-01776-t002:** Formulation and proximate composition of diets.

Ingredients	COM	EXP-S	EXP-I	EXP-A
Fish meal-60 (Thailand)	40	53.5	50.8	53
Soybean meal-46 (Thailand)	36	18	18	16
Casava leaf (Laos)	8	12.4	12	12
Corn (Laos)	7	7	7	6.5
Rice brain (Laos)	6	5.7	5.7	6
Premix (Hungary) ^1^	3	3	3	3
Alga meal (Norway) ^2^	0	0	0	3.5
Black soldier fly meal (Laos)	0	0	3.5	0
Bioactive compound (Austria and France) ^3^	0	0.4	0	0
Proximate composition of the diets (%, as is)				
Water content	9.5	6.7	6.8	6.5
Crude protein	37.0	36.8	36.1	37.5
Crude fat	3.19	3.05	2.66	3.02
Ca	3.06	3.21	3.21	3.21
P	1.73	1.65	1.62	1.72

^1^ Vitafort Zrt. (Dabas, Hungary) with product number 222-374-65; ^2^
*Nannochloropsis gaditana*: produced by NORCE (Norwegian Research Centre) for research purposes; ^3^ Mixture of Yang product (Lallemand, Blagnac, France) and Syrena Boost product (Delacon Biotechnik, Engerwitzdorf, Austria) with 1 g/kg and 200 mg/kg doses, respectively; COM: commercial feed; EXP-S-diet supplemented with bioactive material; EXP-I-diet supplemented with black soldier fly meal; EXP-A-diet supplemented with *N. gaditana*.

**Table 3 animals-15-01776-t003:** Primers used for real-time quantitative PCR.

Target Gene	Forward Primer	Reverse Primer	Efficiency	Reference
*sod*	GACGTGACAACACAGGTTGC	TACAGCCACCGTAACAGCAG	2.02	[25]
*cat*	TCAGCACAGAAGACACAGACA	GACCATTCCTCCACTCCAGAT	2.00	[25]
*gpx*	CCAAGAGAACTGCAAGAACGA	CAGGACACGTCATTCCTACAC	2.03	[25]
*igf-1*	GTTTGTCTGTGGAGAGCGAGG	GAAGCAGCACTCGTCCAGG	2.02	[26]
*tnf-α*	GAGGTCGGCGTGCCAAGA	TGGTTTCCGTCCACAGCGT	2.10	[25]
*il-1β*	TGCTGAGCACAGAATTCCAG	GCTGTGGAGAAGAACCAAGC	1.86	[25]
*il-8*	GCACTGCCCGCTGCATTAAG	GCAGTGGGAGTTGGGAAGAA	2.03	[25]
*ifn-γ*	TGACCACATCGTTCAGAGCA	GGCGACCTTTAGCCTTTGT	2.01	[25]
*18S*	GTGCATGGCCGTTCTTAGTT	CTCAATCTCGTGTGGCTGAA	1.98	[27]

*sod:* superoxide dismutase; cat: catalase; *gpx*: glutathione peroxidase; *igf-1*: insulin-like growth factor 1; *tnf-α*: tumor necrosis factor alpha; *il-1β:* interleukin 1 beta; *il-8*: interleukin-8; *ifn-γ*: interferon gamma; *18S*: 18S rRNA.

**Table 4 animals-15-01776-t004:** Growth performance of Nile tilapia (n = 240 fish/group, mean ± SD).

Treatments	COM	EXP-S	EXP-I	EXP-A	*p*-Value
**IBW (g)**	147.8 ± 2.1	144.0 ± 1.4	140.2 ± 15.8	142.5 ± 2.1	0.863
**FBW (g)**	337.5 ± 68.6	344.5 ± 21.9	344.0 ± 28.3	348.0 ± 46.7	0.994
**SGR (gday^−1^)**	1.67 ± 0.30	1.78 ± 0.11	1.84 ± 0.1	1.81 ± 0.03	0.845
**WG (g)**	190.0 ± 59.4	200.5 ± 20.5	204 ± 12.7	205.5 ± 48.8	0.489
**FCR (gg^−1^)**	1.57 ± 0.54	1.38 ± 0.04	1.29 ± 0.15	1.45 ± 0.49	0.875
**SR (%)**	94.6 ± 0.6	96.3 ± 1.8	96.3 ± 1.8	92.9 ± 0.6	0.150
**CF (gcm^−3^)**	3.80 ± 0.32	3.87 ± 0.33	4.27 ± 0.02	3.99 ± 0.32	0.435
**Feed Cost (€/kg)**	0.77	0.88	0.95	0.95	NR
**Specific Feed Cost** **(€/kg fish)**	1.21 ± 0.15	1.21 ± 0.03	1.23 ± 0.07	1.26 ± 0.09	0.547

IBW: initial average body weight; FBW: average final body weight; WG: weight gain; SGR: specific growth rate; FCR: feed conversion ratio; SR: survival rate; CF: condition factor; COM: commercial feed; EXP-S: diet supplemented with bioactive material; EXP-I: diet supplemented with black soldier fly meal; EXP-A: diet supplemented with *N. gaditana;* NR: not relevant.

## Data Availability

The original contributions presented in this study are included in the article. Further inquiries can be directed to the corresponding author.

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
