# Peer review of "Effects of Feed Additives (Nannochloropsis gaditana and Hermetia illucens) on Growth and Expression of Antioxidant and Cytokine Genes in Nile Tilapia (Oreochromis niloticus) Subjected to Air Exposure Stress"

_animals, 2025, doi:10.3390/ani15121776_

Round 1

Reviewer 1 Report

Comments and Suggestions for Authors

The MS "Functional feeds for healthy fish: towards immunization of Nile tilapia (Oreochromis niloticus) in Laos"

Italicize the scientific names wherever it appears in the manuscript. 

Author needs to specify "different feed additives" in the abstract section. There is no clarity which feed additive they are using.

What do you mean by functional feed additive. There is no mention about it in abstract.

You need to specify " As control a commercially available feed for tilapia was selected" which company feed you have used in the experiment. Also, you need to give their ingredient composition and proximate composition. 

What do you mean by "air-exposure stress"?

What do you mean by "adequate fish growth"?

Line 34-35, rephrase these lines.

This is vague statement "Based on our results, we recommend the inclusion of any of the studied additives to help protect fish from unexpected air exposure during the farming period". Author needs to be very specific to specify the concluding statement. 

Specifying country in the title of the manuscript does not suits these kinds of studies. Authors are suggested to remove the same from the title.

Line 42-48 should be removed from the manuscript and authors are suggested to give introduction about functional feed or related literature.

Line 51-63, short it to within 2-3 sentences.

Which commercially available feed additive you have used in your study. Please specify. 

The introduction section is written poorly without much emphasis on why it was required to test different feed additives in the present study. 

There is no clarity in figure 1. Author needs to write the different abbreviation in the footnote.

Crude fat is reasonably low in the feed. Could you please explain why such low amount of crude fat is there in the diet? Please refer to the lipid requirement of species. 

Try to combine fig 1 and fig. 2.

Material and methods section is written well.

Results and discussion section is also written well. 

Conclusion section is very elaborated. Authors are suggested to shorten this section to make it short and meaningful. 

Author Response

Please find the response in the attached file.

Reviewer 2 Report

Comments and Suggestions for Authors

As extra feed additives, add extra costs to the administered diets, a cost-benefit analysis should be presented to substantiate their use in feed for Nile Tilapia

Author Response

Please find the response to the review in attachment.

Reviewer 3 Report

Comments and Suggestions for Authors

This study explores the effects of different feed additives (Nannochlorophsis gaditana, black soldier fly larvae meal, and commercial additives) on the growth, antioxidant capacity, and immune gene expression of Nile tilapia, providing valuable references for the development of functional feeds and holding certain practical application value. The overall experimental design is rational, and the data support some of the conclusions. However, further improvements are still needed in the details of the methods, interpretation of the results, depth of the discussion, and language expression. The specific issues are as follows:

Line 2-3: It is recommended to specify the specific types of additives in the title.

Line 16: “Oreochromis niloticus” should be in italics, and the same applies to the other Latin names.

Line 33: The notation of p<0.05 here is incorrect. The original text means that there is no significant difference, which should be p>0.05. Moreover, p-values are generally not required in the abstract.

Line 38-39: The statement " we recommend the inclusion of any of the studied additives " in the conclusion is contradictory to the significant advantages of the EXP-A group in the results. It should clearly prioritize the recommendation of Nannochlorophsis gaditana.

Line 80: Please confirm whether a comma is needed between “European seabass [16] and “yellow catfish (Pelteobagrus fulvidraco).”

Line 118-119: The dissolved oxygen saturation in the water quality parameters is relatively low (41.5-66.6%), which may interfere with the stress response and metabolism of the fish. It is necessary to discuss the potential impact of this condition on the experimental results.

Line 128: It is necessary to supplement the name of the ethics review institution and the approval number.

Line 295: Compared with "isoenergetics", "isoenergetic" is more appropriate and commonly used.

Line 306-308: It is necessary to conduct an in-depth explanation of why only the EXP - A group significantly upregulated antioxidant genes while the effects in other groups were not significant (such as the mechanism of action of specific components in Nannochlorophsis gaditana).

Line 420-421: The original suggestion of "inclusion of any of the studied additives" contradicts the fact that the EXP-A group significantly outperforms the other groups in the data. It is recommended to be revised such as: "We recommend giving priority to supplementing 3.5% of N. gaditana to enhance the antioxidant defense capacity, and meanwhile, further studies are required to optimize the dosages of BSFL and commercial additives."

In addition, it is a rather important issue that the expression of immune-related genes is not presented in the results section of the manuscript. Please check whether there is any omission of results.

Author Response

Please find the response to the review in the attached file.

Round 2

Reviewer 1 Report

Comments and Suggestions for Authors

The manuscript is substantially revised, and authors have addressed all the query satisfactorily. I have no further query. 

Author Response

Thank you for your comments.

Reviewer 2 Report

Comments and Suggestions for Authors

Not all issues which have been pinpointed in the initial manuscript, have finally been addressed by the authors. 

The issues which have not been addressed, are:

3.2. Lines 171-175: regarding the stress trial, why have authors chosen “five minutes” as the interval time for the air exposure, and not two, three, four, or six minutes ? Is there a referred specific protocol for this? As fish have been returned back to the water after the air-exposure, have fish survived after this “5-minute air exposure”? 

3.3. Lines 100-112: The Initial and Final Stocking Density of fish in the cages, are somehow low (< 1 kg/m-3 ). This does not correspond to real commercial fish farming conditions, where stocking densities range between 10-20  kg/m-3.  However, this is not a crucial drawback for the experimental design and credibility of the experimental outcome, BUT the authors have to justify this methodology. Why have they chosen such a low density? Is there a specific reason ? 

3.4. Lines 144-145 (Table 2): There is a drawback (in the experimental design), as far as the protein and amino acid content of the experimental feeds, is concerned: the COM diet contains 40% fishmeal and 36% soybean meal. In the rest of experimental diets, soybean meal is diminished by half (18% of the diet) and fishmeal is increased by 10-13%. Does this increase in fishmeal, compensate for the loss of amino acids supplied by soybean meal ? (which has been halved compared to the COM diet). This is clearly acknowledged in the Discussion Section (lines 296-301), and is good that the authors realize the negative side-effect of this specific experimental drawback.

3.5. Lines 430-431: "However, the immunomodulatory effect of the additives observed after the feeding period was negated by the air exposure stress in all treatments". = this statement overturns almost all the positive results of the experiment 

Author Response

Please see in the attachment.

Reviewer 3 Report

Comments and Suggestions for Authors

The author has made considerable revisions to the MS and it is acceptable for publication.

Author Response

Thank you for the comments.